

# Predicting sport event outcomes using deep learning

Jianxiong Gao[1], Yi Cheng[1] and Jianwei Gao[2]

[1] Department of Physical Education, Chengdu University of Information Technology, Chengdu, China
[2] School of International Education, Sichuan University of Media and Communications, Chengdu, China

## ABSTRACT

Predicting the outcomes of sports events is inherently difficult due to the unpredictable nature of gameplay and the complex interplay of numerous influencing factors. In this study, we present a deep learning framework that combines a one-dimensional convolutional neural network (1D CNN) with a Transformer architecture to improve prediction accuracy. The 1D CNN effectively captures local spatial patterns in structured match data, while the Transformer leverages self-attention mechanisms to model long-range dependencies. This hybrid design enables the model to uncover nuanced feature interactions critical to outcome prediction. We evaluate our approach on a benchmark sports dataset, where it outperforms traditional machine learning methods and standard deep learning models in both accuracy and robustness. Our results demonstrate the promise of integrating convolutional and attention-based mechanisms for enhanced performance in sports analytics and predictive modeling.

# INTRODUCTION

Predicting the outcomes of sports events has become an expanded area of study and application with the improvement of data analysis and machine learning. In addition to being interesting, accuracy of prediction is also useful for coaches, players, and sports organizations. With data analysis, coaches can optimize game strategies for matches, enhance players' performance, and guide the selection of the lineup and strategy. This multi-aspect sports prediction encompasses many variables, including player fitness, weather, and tactics, all of which contribute to the uncertainty of sporting outcomes.

The determinants of the result of a sports match are numerous and interrelated. Player condition is an important determinant since players' performance levels of players may shift due to numerous factors such as injury, tiredness, and mental reasons (*Huang & Li, 2021*). Moreover, team strategies can influence match results. For instance, formation choice, playing style, and game adaptation are all variables that may influence the success of a team's strategy (*Tadesse et al., 2022*). Coaches must therefore have to be cautious and adaptable, utilizing predictive analytics in forecasting how these factors will affect the performance of their team. Weather is also a factor that determines sporting events. Studies have indicated that poor weather can influence player performance, especially in

Corresponding author
Jianwei Gao, vickykao163@163.com

outdoor sports like soccer and cricket (*Sankaranarayanan, Sattar & Lakshmanan, 2014*). The environmental conditions should be considered by analysts and coaches in planning games and making lineup decisions. For instance, heavy rain may demand a defensive approach by maintaining control over the ball without taking unnecessary risks. By integrating weather conditions into model forecasts, teams can better plan for the issues of changing conditions. Beyond these climatic factors, sports performance must also be addressed on psychological aspects. The mental state of athletes can greatly contribute to their level of performance, with stress, motivation, and confidence being principal factors (*Malloy, Kavussanu & Yukhymenko-Lescroart, 2022*). Coaches can utilize predictive analytics to analyze the mental readiness of their players, then schedule preparation and training based on the findings. By having a general overview of their players, they can generate improved performance outcomes and greater overall team success.

One of the most primary applications of sports outcome prediction is in the field of coaching. Coaches can utilize predictive models to assess the potential performance of their teams under different conditions. For instance, *Wilkens (2021)* emphasizes the role of machine learning techniques, in modeling sports outcomes, with a particular application being tennis, where players' fitness varies significantly on a weekly basis. This adaptability allows coaches to develop individualized plans based on predicted player performance and optimize their chances of winning in competitions. Furthermore, the integration of contextual factors, such as the psychological state of athletes and their interpersonal dynamics, can enhance the understanding of team performance (*Tadesse et al., 2022*). Optimization of competition strategies based on data is not limited to use in individual sports. For team sports, it is essential to know the interaction among players and how outside factors, such as weather and game location, affect the game. Research indicates that various parameters, including game location and refereeing can significantly influence game outcomes (*Bunker & Susnjak, 2022*). With the application of advanced statistical methods and machine learning algorithms, coaches are able to model such variables in order to develop better game strategies. For example, the utilization of fuzzy logic-based models has been shown to improve predictive accuracy by addressing the complexity and uncertainty of sporting data (*Liu, 2024*).

The development of machine learning (ML) in sports outcome prediction has evolved into a revolutionary force, transforming how analysts and teams approach performance analysis and strategy formulation (*Bunker & Susnjak, 2022*). The integration of big data and artificial intelligence (AI) in sports analytics has enabled the processing of vast amounts of data, allowing for more accurate predictions and insights into player performance, team cohesion, and match outcomes (*Beal, Norman & Ramchurn, 2019*). Traditional methods, such as expert analysis and basic statistical techniques, have increasingly been supplemented or replaced by sophisticated ML models that leverage complex algorithms to analyze patterns and make predictions based on historical data. The role of big data in sports analytics cannot be overstated. With the advent of advanced tracking technologies, teams now have access to more information than ever, ranging from player movement to in-game statistics and even biometric data. This wealth of data provides a fertile ground for machine learning applications, which can identify trends and

correlations that are not as apparent when using traditional methods of analysis. For instance, *Soto Valero (2016)* showed that ML techniques can outperform conventional statistical methods in predicting outcomes in various sports, including baseball and basketball, by utilizing comprehensive datasets that encompass multiple variables influencing game results.

Although traditional machine learning methods, such as decision trees and boosting algorithms, have shown promise in sports event forecasting, they often struggle to capture the intricate dependencies and sequential nature of sports data. These limitations necessitate the exploration of more advanced deep learning techniques capable of effectively encoding rich relationships in structured data. Recent advances in deep learning, particularly the emergence of Transformer-based architectures (*Vaswani et al., 2017*), have revolutionized many fields, including natural language processing and time-series forecasting. In this study, we leverage the power of Transformers and 1D convolutional neural networks (1D CNN) to enhance the accuracy of sports event outcome prediction. The self-attention mechanism of the Transformer enables it to capture long-range dependencies and complex feature interactions, while 1D CNN efficiently processes structured tabular data by extracting crucial spatial and temporal patterns from the dataset. By combining these architectures, our approach aims to address the limitations of traditional methods and provide a robust framework for sports prediction. This research seeks to bridge the gap between state-of-the-art deep learning models and practical applications in sports analytics, offering valuable insights for analysts, teams, and stakeholders.

## RELATED WORK

Traditional methods of predicting sport outcomes have a high reliance on rating systems and statistical models. One of the most widely used techniques is the Elo rating system (*Elo, 1966*), which was originally developed for ranking chess players and was later on extended to other types of sports such as football and basketball. The Elo system updates team or player ratings based on game results and makes a probabilistic prediction of future game outcomes. In 2010, *Hvattum & Arntzen (2010)* imporeved the basic Elo model including modifications that incorporate margin of victory and home advantage for predicting football result. Another common approach involves regression-based models, such as Poisson regression, *Dixon & Coles (1997)* introduced an adjustment to the Poisson model to better capture low-scoring events in football. In 2010, *Baio & Blangiardo (2010)* explored to incorporate prior knowledge and uncertainty in predictions by using Bayesian approaches. Moreover, Markov Chain (*Frigessi & Heidergott, 2011*) models have been applied to model the progression of matches over time and improve predictive performance (*Koopman & Lit, 2012*). Other studies have explored hybrid approaches, for example, *Lasek, Szlávik & Bhulai (2013)* combined Elo ratings with Bayesian models to enhance prediction accuracy. Statistical models, while interpretable and effective, often struggle with capturing complex interactions between variables and adapting to dynamic changes in team performance, which has led to the adoption of machine learning-based approaches.

Deep learning algorithms have been recently employed with remarkable success in making sports outcome predictions. For instance, convolution neural networks (CNNs) have been employed to analyze video recordings to check for movement detection and performance evaluation (*Cust et al., 2018*; *Moodley, Van der Haar & Noorbhai, 2022*). The models can automatically discover features from raw data, thus reducing the need for manual feature engineering. Experiments have verified the potential of deep learning models in accurately predicting the results of games such as basketball and baseball (*Huang & Li, 2021*; *Atta Mills et al., 2024*). These advancements in technology enable coaches and analysts to process large amounts of data, uncovering patterns and insights that can inform decision-making processes. The interplay between various factors that affect the result of sports shows the intricacy of prediction in this field. As highlighted by *Tadesse et al. (2022)*, psychological needs fulfillment as well as contextual variables play crucial roles in shaping the developmental outcome of sports players. This suggests that a comprehensive understanding of the multifaceted nature of sports performance is essential for effective prediction. By taking into account a wide range of factors, including morale of players, team cohesion, and conditions of the environment, coaches can optimize their predictive power and optimize their teams' potential for achievement. Additionally, ensemble methods that combine multiple algorithms have shown promise in improving prediction accuracy by leveraging the strengths of different models (*Bunker & Thabtah, 2019*; *Imbach et al., 2022*). *Sun (2022)* explored the use of chaos theory in conjunction with machine learning to enhance prediction models by analyzing the dynamic nature of sports performance. Machine learning models can utilize past injury information as well as performance records to calculate risk factors and predict potential injuries, allowing teams to implement preventive measures (*Li et al., 2023*). This method is not only safe for athletes but also enhances training programs as well as match planning. The integration of machine learning into sports analytics has also led to the development of smart sports platforms that utilize real-time data for decision-making. These are based on cloud computing and mobile technology to provide coaches and analysts with real-time insights during the game, so they can effect real-time changes in strategy (*Gong, 2023*). The ability to analyze data in real-time is a significant advantage over traditional methods, which often rely on post-game analysis.

# MATERIALS AND METHODS

## Dataset

The European Soccer Database from Kaggle is a comprehensive dataset that provides detailed information on European football matches, teams, and players, making it a valuable resource for predictive modeling and sports analytics. It includes data from 11 European countries across seasons 2008 to 2016, covering over 25,000 matches and 10,000 players. This dataset includes some key features:

- Match Data: includes match results, team formations, player line-ups, and detailed in-game events such as goals, assists, fouls, and possession statistics.

- Player & Team Attributes: extracted from EA Sports' FIFA series, offering insights into player skills, team strengths, and performance trends.
- Betting Odds: collected from up to 10 providers, allowing analysis of market expectations *vs.* actual outcomes.
- Event-Level Match Data: includes goal types, corners, crosses, fouls, and other in-game occurrences for over 10,000 matches.

This dataset can be used to develop machine learning models to predict match results (Win, Draw, Defeat). By leveraging these features, the dataset supports research in sports analytics and team performance evaluation.

### Data preprocessing

To build a match outcome prediction model, data from the European Soccer Database is processed and extracted into key features. This process includes labeling match results, aggregating player information, computing team performance statistics, integrating and combining all these data into a complete training dataset. Initially, each match is labeled based on the number of goals scored by the home team (*home_team_goal*) and the away team (*away_team_goal*). The label represents the home team's result in three possible states:

- **Win**: if the home team's goals exceed the away team's goals.
- **Draw**: if both teams have the same number of goals.
- **Defeat**: if the home team's goals are fewer than the away team's goals.

Next, to assess team quality in each match, FIFA data is used to extract the overall rating (*overall_rating*) of players. For each player, the closest rating before the match date is selected as the representative value. For a given match with a set of players from both teams, player information is extracted using the following formula:

$$\text{rating}(p_i) = \max(\text{overall\_rating}(p_i, d)) \quad \text{with } d < D, \tag{1}$$

where *d* is the match date and represents previous time points. In addition, key statistics from their past matches are analyzed. These include the number of goals scored, goals conceded, and matches won over a specific period. The goal difference is also considered, providing insight into a team's overall strength by comparing goals scored against goals conceded. By aggregating these metrics, we obtain valuable performance indicators that help in predicting future match outcomes. This approach ensures that both offensive and defensive capabilities are factored into the analysis, leading to more accurate predictions.

After extracting individual player ratings, they are aggregated to create a feature set representing the team's lineup quality. Once all features are collected, they are merged to form a complete dataset by combining team performance statistics, head-to-head history, and FIFA player ratings based on the match identifier. After extracting individual player ratings, they are aggregated to create a feature set representing the team's lineup quality. Once all features are collected, they are merged to form a complete dataset by combining
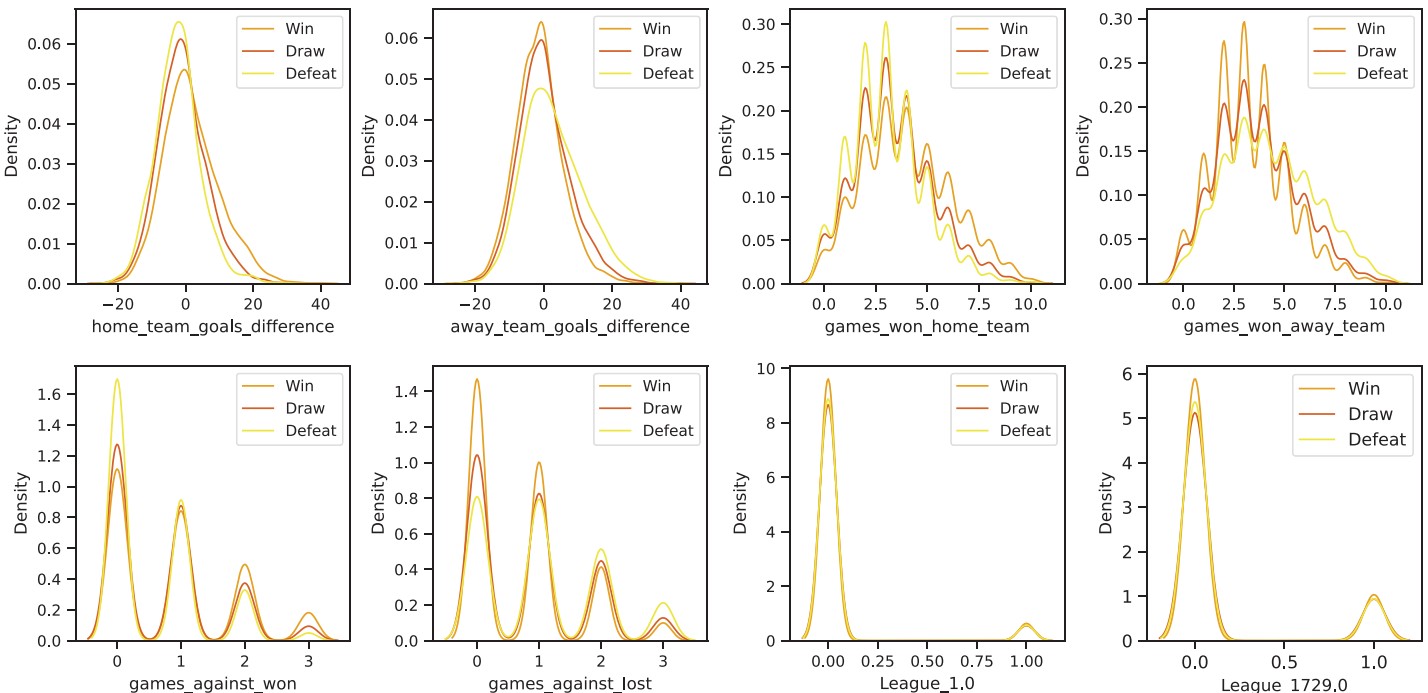

**Figure 1** Kernel density estimation (KDE) plots of selected features in the dataset, grouped by match outcome labels: Win, Draw, and Defeat.

team performance statistics, head-to-head history, and FIFA player ratings based on the match identifier. To effectively process the data for predictive modeling, we apply different encoding techniques for categorical and numerical variables.

- Categorical Feature Processing ($x_{cat}$): categorical variables, such as league identifiers, cannot be directly used in mathematical models as they lack numerical meaning. Therefore, we employ an *embedding layer* to map these categorical values into a lower-dimensional numerical space while preserving semantic relationships among different categories. The use of embeddings not only enhances model interpretability but also mitigates the sparsity issue inherent in categorical variables when fed into predictive models.

- Continuous Feature Processing ($x_{cont}$): continuous variables contain critical numerical information, such as goals scored, shots on target, and ball possession percentage. To ensure that these variables do not introduce scale discrepancies among different features, we apply either normalization or standardization techniques, depending on the distribution of each variable. Additionally, to maintain data integrity and avoid potential biases in the model, we remove rows with missing values rather than using imputation techniques, as filling in missing values (*e.g.*, with mean or median) could introduce distortions in model interpretation. To visualize the distribution of continuous variables based on the target labels (Win, Draw, Defeat), we present the probability density plots using Kernel Density Estimation (KDE). Figure 1 illustrates the differences in feature

**Table 1 Description of the training, validation, and test datasets.**

| Dataset | Duration | Win | Defeat | Draw | Total |
|---------|----------|-----|--------|------|-------|
| Training | 2009-01-03 to 2013-12-29 | 6,105 | 3,707 | 3,343 | **13,155** |
| Validation | 2014-01-01 to 2014-12-28 | 1,305 | 864 | 718 | **2,887** |
| Test | 2015-01-01 to 2015-12-31 | 1,354 | 919 | 779 | **3,052** |
| **Total** | **2009-01-03 to 2015-12-31** | **8,764** | **5,490** | **4,840** | **21,374** |

distributions across different outcome groups, highlighting the relationship between input features and predicted results.

### Data splitting

To ensure that the inherent temporal structure of the time-series data remains intact, we carefully partitioned the dataset into three distinct subsets: a training set, a validation set, and a test set. This approach preserves the chronological order of observations, preventing any potential data leakage from future time points into past observations, which could otherwise compromise the model's ability to generalize to unseen data.

The training set includes a total of 13,155 data samples, systematically collected over a period spanning from January 3, 2009, to December 29, 2013. This subset serves as the foundational dataset for model learning, allowing the predictive model to recognize underlying patterns and relationships within the data. Following this, the validation set consists of 2,887 samples recorded between January 1, 2014, and December 28, 2014. The primary role of this subset is to fine-tune hyperparameters and assess the model's performance on previously unseen data before making final adjustments. It provides an unbiased evaluation of the model's learning progress and helps mitigate the risks of overfitting. Lastly, the test set is composed of 3,052 samples covering the period from January 1, 2015, to December 31, 2015. This dataset remains completely separate from the training and validation phases and is utilized exclusively for evaluating the final model's predictive performance in real-world scenarios.

In total, the dataset spans nearly seven years, from January 3, 2009, to December 31, 2015, with a cumulative count of 21,374 samples. A detailed breakdown of the data distribution across these subsets is provided in Table 1, offering a comprehensive overview of the dataset division for training, validation, and testing purposes.

## Model architecture

### Model selection rationale

In this study, we introduce a hybrid deep learning model that integrates Transformer layers with a residual one-dimensional convolutional neural network (1D CNN) for forecasting sports event outcomes. As shown in Fig. 2, continuous features are processed through a 1D CNN block to capture local patterns, while categorical features are encoded *via* column embeddings to generate learnable parametric representations. These embeddings are then passed through Transformer layers (*Vaswani et al., 2017*) to model

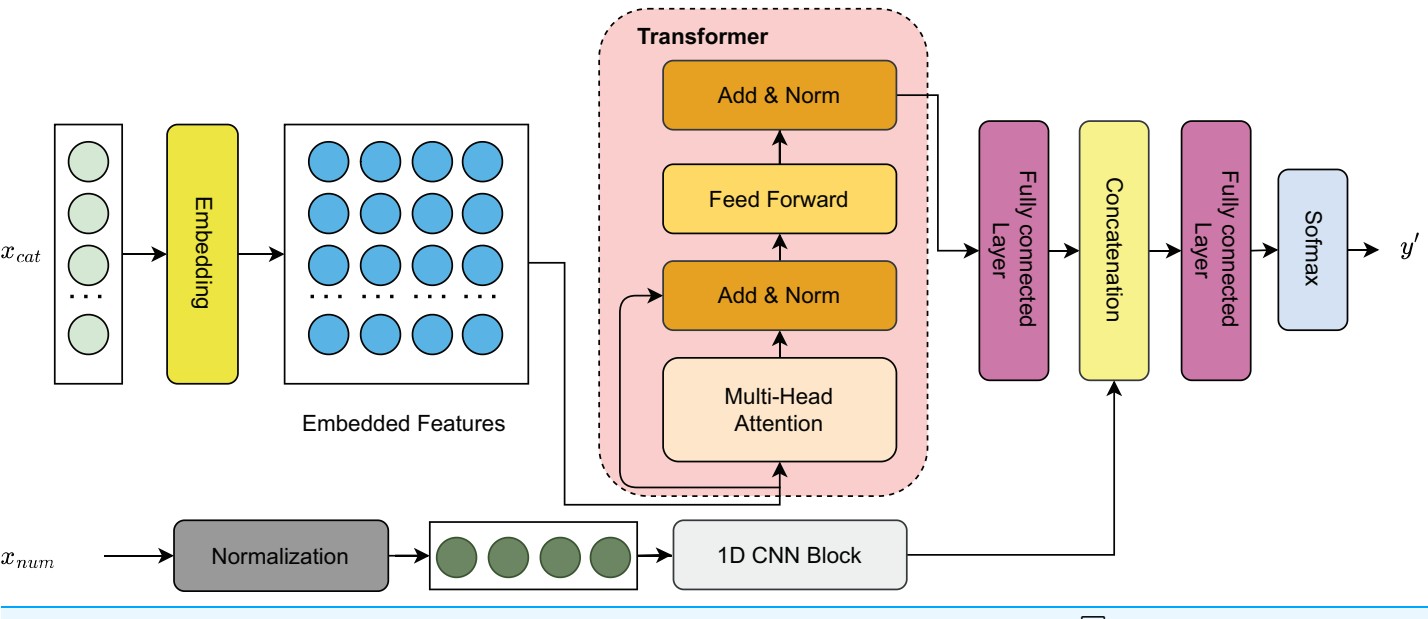

**Figure 2** **Visualization of the model architecture.**

long-range dependencies. The resulting representations from both feature types are concatenated and passed through a fully connected (FC) layer for dimensionality reduction, followed by a softmax activation to produce the final outcome predictions.

The selection of the hybrid model architecture combining a residual 1D CNN with Transformer layers was guided by the complementary strengths of these techniques in handling structured sequential data. The 1D CNN was chosen for its ability to effectively capture local spatial patterns and short-term dependencies within continuous features, which are common in sports event data. Meanwhile, the Transformer architecture was selected for its superior capacity to model long-range interactions through self-attention mechanisms, making it well-suited for capturing complex contextual relationships among categorical and embedded features. This combination enables the model to learn both low-level and high-level feature interactions, which are critical for accurate outcome prediction. Alternative approaches, including standalone CNNs, recurrent neural networks (RNNs), and traditional machine learning models, were initially considered and evaluated in preliminary experiments. However, they demonstrated inferior performance in capturing global dependencies or exhibited limitations in scalability. Therefore, the proposed hybrid design was selected for its balance of expressiveness, interpretability, and empirical performance.

### Data encoding

Before being forwarded into the model, both categorical and continuous features need preprocessing steps that include encoding and normalization to ensure the data is in a suitable format for effective learning.

Specifically, categorical features are transformed into dense vector representations through learnable embedding layers. Let the set of categorical features be denoted as

$\{c_1, c_2, \ldots, c_n\}$, where each categorical feature $c_i$ assumes discrete values from a predefined vocabulary $V_i$. The vocabulary $V_i$ contains all possible unique values that the feature $c_i$ can take, and its cardinality is represented by $|V_i|$. For each categorical feature $c_i$, an embedding matrix $E_i \in \mathbb{R}^{|V_i| \times d}$ is defined, where $d$ is the dimension of the dense embedding vectors. This matrix $E_i$ is learnable and can be optimized during the model training.

For continuous features, we apply only a normalization layer to standardize the input data. This normalization step is crucial to ensure that the range and distribution of continuous features do not negatively impact the model training. After normalization, the continuous features are directly fed into 1D CNN, which can effectively learn from the standardized continuous inputs.

### Transformer

Transformer is a groundbreaking neural network architecture that has fundamentally reshaped the landscape of natural language processing (NLP) and deep learning. Transformers leverage self-attention mechanisms to capture long-range dependencies in data efficiently. This architecture eliminates the need for recurrent computations, enabling parallelization and significantly improving training efficiency. At the core of the Transformer model is the self-attention mechanism, which computes contextualized representations of input tokens based on their relationships with all other tokens in a sequence. Given an input sequence represented as a matrix $X$, self-attention is computed as:

$$\text{Attention}(Q, K, V) = \text{softmax}\left(\frac{QK^T}{\sqrt{d_k}}\right)V, \tag{2}$$

where $Q$, $K$, $V$ are query, key, and value matrices derived from $X$, and $d_k$ is the dimension of the key vectors. The multi-head attention mechanism extends this by applying multiple attention heads in parallel:

$$\text{MultiHead}(Q, K, V) = \text{Concat}(\text{head}_1, \ldots, \text{head}_h)W^O, \tag{3}$$

where each attention head computes independent self-attention, followed by a linear transformation with weight matrix $W^O$. The Transformer also incorporates position-wise feedforward layers and layer normalization to stabilize training.

### 1D CNN block

The 1D CNN layer is an effective architecture for processing one-dimensional sequential data, such as time series, sensor readings, and speech signals. 1D CNN applies convolutional filters across the temporal axis, capturing local dependencies and feature patterns efficiently. The key operation in a 1D CNN layer is the 1D convolution, expressed as:

$$y_i = \sum_{j=0}^{k-1} w_j \cdot x_{i+j}, \tag{4}$$

where $x$ is the input sequence, $w$ represents learnable filter weights, $k$ is the filter size, and $y$ is the output feature map.

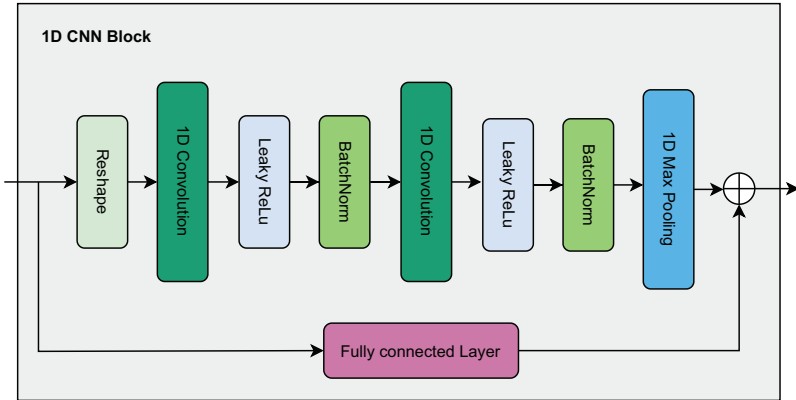

**Figure 3 Visualization of the 1D CNN block.**

In this paper, due to continuous features being represented as a one-dimensional vector, we design a residual network architecture combined with a 1D CNN as shown in Fig. 3. First, the continuous features are reshaped to serve as input to the 1D CNN. Here, the data passes through two consecutive 1D CNN layers, each followed by a Leaky Rectified Linear Unit (ReLU) activation function and Batch Normalization to enhance stability and convergence speed during training. After passing through the 1D CNN layers, the data is fed into a Max Pooling layer to reduce dimensionality and is finally flattened into a one-dimensional vector.

Parallel to the main branch, the residual branch helps preserve the original information by passing the input data through an FC layer to ensure it has the same dimensions as the output of the main branch. Then, the output from the residual branch is directly added to the output of the main branch to produce the final output. The residual connection acts as a shortcut mechanism, allowing the original information to be transmitted to later layers without losing important features, thereby mitigating the gradient vanishing problem as the network becomes deeper.

## Evaluation metrics

To evaluate the performance of our models, multiple assessment metrics were calculated, including Accuracy (ACC) with threshold of 0.5, Weighted Recall, Weighted Precision, Matthews Correlation Coefficient (MCC), and the Weighted F1-score. These metrics are mathematically defined as follows:

$$\text{Weighted Recall} = \sum_i w_i \times \frac{TP_i}{TP_i + FN_i}, \tag{5}$$

$$\text{Weighted Precision} = \sum_i w_i \times \frac{TP_i}{TP_i + FP_i}, \tag{6}$$

$$\text{Weighted F1} = \sum_i w_i \times \frac{2 \times \text{Precision}_i \times \text{Recall}_i}{\text{Precision}_i + \text{Recall}_i}, \tag{7}$$

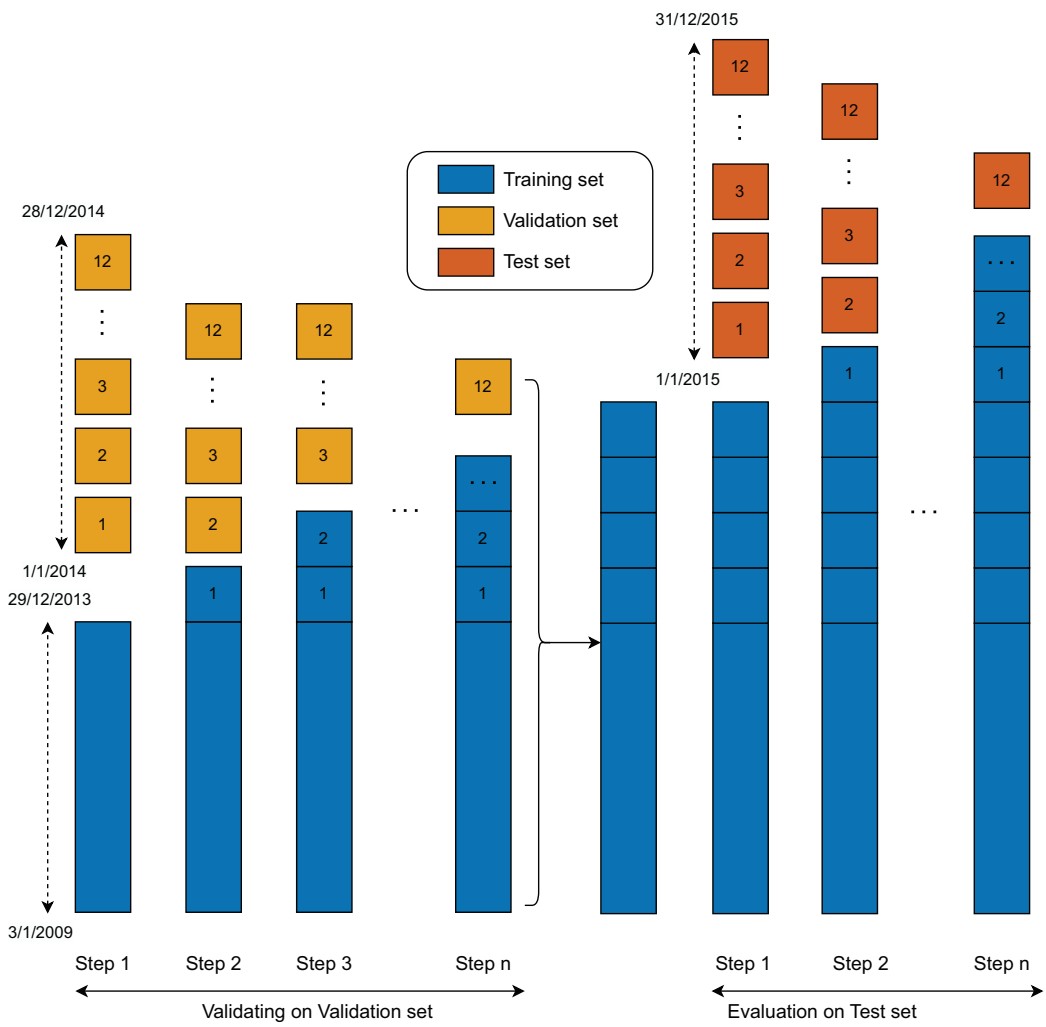

**Figure 4 Validation and test set evaluation workflow.**

$$\text{MCC} = \frac{(TP \times TN) - (FP \times FN)}{\sqrt{(TP + FP)(TP + FN)(TN + FP)(TN + FN)}}, \tag{8}$$

$$\text{ACC} = \frac{TP + TN}{TP + TN + FP + FN}, \tag{9}$$

where *TP* (true positives) represents correctly predicted positive instances, *TN* (true negatives) denotes correctly predicted negative instances, *FP* (false positives) corresponds to incorrectly predicted positive cases, and *FN* (false negatives) accounts for incorrectly predicted negative cases.

## Model training and evaluation

The training procedure was designed to enable all models to learn from the data in a sequential manner (Fig. 4). Initially, each model was trained on the full training set and validated using data from the first month. In Step 1, the first month's data was incorporated into the training set, and the models were re-trained on this updated dataset.

This process was repeated iteratively, with each subsequent month's data added to the training set and used to validate the model, continuing up to Month 12. As a result, the models were re-trained a total of 12 times during the training and validation phase.

Following this phase, a similar training and evaluation procedure was applied to the test set. In this second stage, the models were again created, updated, and evaluated using the same rolling window approach. Since the test set comprises data from the final 12 months, the models were updated and evaluated 12 times accordingly.

The duration for one training epoch is about 152 s. In the training and validating process, the model (Step 1) was trained for over 50 epochs at a learning rate of 0.0005. The following updated models were trained using three additional epochs. At the training and evaluation state, the original model was retrained with a second training set of all training and validation samples. After the first models were obtained, the following updated models were retrained with one additional epoch and tested with the data of the following month. The training process in the second phase was stopped after 12 steps.

## Design of experiments for model benchmarking
### Computing infrastructure
All experiments in this study were conducted on a workstation equipped with an NVIDIA GeForce RTX 3070 GPU with 12 GB of GDDR6 VRAM, which enabled efficient training of deep learning models. The system also featured an AMD Ryzen 7 5800X processor with eight cores and 16 threads, supported by 32 GB of DDR4 RAM running at 3,200 MHz, ensuring smooth multitasking and data processing. A 1TB NVMe SSD was used for high-speed storage and rapid data access. The experiments were performed in a Windows 11 environment, which provided a stable and compatible platform for running the necessary machine learning libraries and frameworks.

### Performance comparison with machine learning and deep lerning models
To evaluate the performance of our proposed model in predicting sports event outcomes, we establish a comparative analysis with traditional machine learning and deep learning models. As baseline machine learning approaches, we consider decision trees (*Rokach & Maimon, 2005*), random forests (*Breiman, 2001*), *k*-nearest neighbors (*Kramer, 2013*), gradient boosting (*Friedman, 2001*), XGBoost (*Chen & Guestrin, 2016*), and CatBoost (*Dorogush, Ershov & Gulin, 2018*).

In addition, deep learning models are explored due to their ability to learn complex patterns within data. We compare our approach against state-of-the-art architectures, including multi-layer perceptrons (MLP) (*Taud & Mas, 2017*), recurrent neural networks (RNNs) (*Marhon, Cameron & Kremer, 2013*), long short-term memory (LSTM) (*Hochreiter & Schmidhuber, 1997*), TabNet (*Arik & Pfister, 2019*), TabTransformer (*Huang et al., 2020*), and TabPFN (*Hollmann et al., 2022*). This experimental setup allows us to assess the effectiveness of our model in capturing key predictive features while addressing challenges such as data scarcity and overfitting.
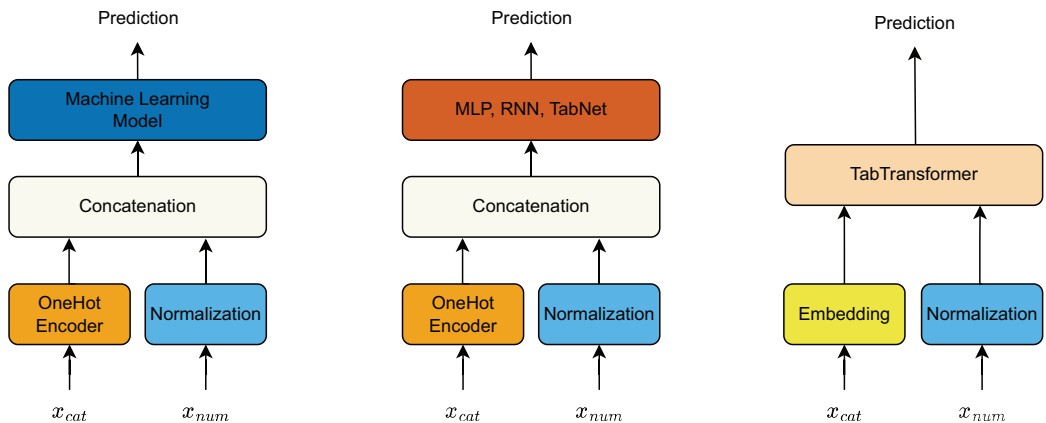

**Figure 5 Preprocessing pipeline for categorical and continuous features across different model types.**

### Input preprocessing for different models

The input features are divided into categorical features and continuous features, with different preprocessing techniques applied depending on the model type (Fig. 5).

- For traditional machine learning models, categorical features are transformed using one-hot encoding, while continuous features are normalized to maintain numerical stability. The processed categorical and continuous features are then concatenated and fed into the machine learning model for prediction.
- Similarly, deep learning models such as MLP, RNN, and TabNet follow the same preprocessing approach as traditional machine learning models. The categorical features undergo one-hot encoding, and continuous features are normalized before being concatenated and used as input to the deep learning architectures.
- In contrast, TabTransformer processes categorical and continuous features differently. Categorical features are passed through an embedding layer to transform them into dense vector representations. Meanwhile, continuous features are normalized. Both processed categorical and continuous feature representations are then input into TabTransformer, which leverages attention mechanisms to model interactions between features effectively.

By standardizing the input preprocessing techniques across different models, we ensure a fair and meaningful comparison of their predictive performance.

### Hyperparameter tuning

To optimize machine learning model performance, hyperparameter tuning is essential. Grid search is used to explore various parameter combinations and determine the best set of hyperparameters for each model. Table 2 summarizes the key hyperparameters tuned for each model. By systematically evaluating different hyperparameter values using grid search, the models can achieve better predictive accuracy and generalization, ultimately improving the reliability of sports event outcome predictions.

**Table 2 Hyperparameter tuning details for machine learning models.**

| Algorithm | Parameter | Range of values |
|---|---|---|
| Decision tree | criterion | 'gini', 'entropy' |
| | max_depth | 6, 8, 10 |
| Random forest | min_samples_leaf | 5, 8 |
| | max_depth | 6, 8, 10 |
| k-Nearest neighbor | n_neighbors | 3, 5, 7 |
| | max_depth | 5, 7, 10 |
| Gradient boost | min_samples_leaf | 5, 8 |
| | max_depth | 5, 7, 10 |
| XGBoost | learning_rate | 0.01, 0.05, 0.1 |
| CatBoost | learning_rate | 0.01, 0.05, 0.1 |
| | depth | 5, 7, 9 |

**Table 3 Hyperparameter tuning details for deep learning models.**

| Model | Parameter | Range of values |
|---|---|---|
| MLP | hidden_size | 64, 128, 256 |
| | num_layers | 3, 4 |
| | learning_rate | 0.001, 0.005, 0.01 |
| | batch_size | 64, 128 |
| RNN | hidden_size | 64, 128 |
| | num_layers | 3, 4 |
| | learning_rate | 0.001, 0.005, 0.01 |
| | batch_size | 64, 128 |
| LSTM | hidden_size | 64, 128 |
| | num_layers | 3, 4 |
| | learning_rate | 0.001, 0.005, 0.01 |
| | batch_size | 64, 128 |
| TabNet | n_steps | 3, 5, 7 |
| | learning_rate | 0.001, 0.005 |
| | batch_size | 64, 128 |
| TabTransformer | transformer_layers | 2, 4 |
| | attention_heads | 4, 6 |
| | d_model | 64, 128 |
| | learning_rate | 0.001, 0.005 |
| TabPFN | softmax_temperature | 0.9, 0.5 |
| | balance_probabilities | True, False |

Next, to ensure fair and competitive performance among the deep learning models, we conducted hyperparameter tuning by adjusting several key parameters for each deep learning architecture as shown in Table 3. These parameters significantly influence convergence speed, stability, and overall model performance.

### Ablation study on model component contributions

To investigate the effectiveness of each component in our proposed hybrid model, we conducted an ablation study by systematically modifying or removing key modules and analyzing their impact on performance. The study examines three primary aspects: (1) the Transformer component, (2) the residual 1D CNN block, and (3) categorical feature embeddings. For the Transformer module, we replaced it with a fully connected network that directly processes categorical embeddings. To assess the role of the residual 1D CNN block, we substituted it with an MLP layer and removed the residual connections. To evaluate categorical embeddings, we replaced them with one-hot encoded vectors and measured changes in model complexity and predictive performance. The results of the ablation study across multiple experiments show variations in model performance under different configurations, providing insights into the contribution of each component.

## RESULTS AND DISCUSSION

### Comparison results with machine learning models

As shown in Table 4, our proposed model outperformed all baseline approaches in result prediction of sports events. Specifically, it achieved the best accuracy of 54.73% and MCC of 0.2666, demonstrating its ability to distinctly capture major patterns in the data and enhance generalization.

Tree-based models exhibited varying performance levels. The decision tree model, prone to overfitting, recorded the lowest accuracy among tree-based methods at 41.31% with an MCC of 0.0910. In comparison, random forest improved upon this, achieving an accuracy of 50.20% and an MCC of 0.1875. The ensemble nature of random forest mitigates overfitting and enhances model generalization; however, its limitations in capturing strong feature relationships hinder further performance gains. Likewise, the $k$-nearest neighbors model performed similarly to the decision tree, with an ACC of 40.65% and an MCC of 0.0790, likely due to its sensitivity to hyperparameters such as $k$ and distance metrics, as well as its tendency to overfit in pre-match prediction scenarios.

Boosting techniques provided notable performance enhancements over conventional methods. Gradient boosting, for instance, showed significant improvement, achieving an ACC of 53.76% and an MCC of 0.2482, underscoring the effectiveness of iterative learning in identifying complex data patterns. XGBoost and CatBoost, both advanced boosting frameworks, demonstrated strong results, with XGBoost reaching an ACC of 51.02% and an MCC of 0.2006, whereas CatBoost slightly surpassed it with an ACC of 52.54% and an MCC of 0.2263. CatBoost's superior performance can be attributed to its efficient handling of categorical features, which plays a crucial role in sports event prediction.

Overall, our proposed method consistently outperformed all prior approaches, securing the highest ACC and MCC. The improved precision and recall metrics show the model's capability in maintaining the essential predictive features. These outcomes highlight the capability of ensemble and boosting methods in addressing the challenges of sports event prediction. Nevertheless, the moderate accuracy levels across all models suggest that predicting sports events remains a challenging task due to inherent uncertainties and

**Table 4 Test set performance comparison with machine learning models.** Best results are shown in bold.

| Model | ACC | MCC | Weighted recall | Weighted precision | Weighted F1 |
|---|---|---|---|---|---|
| Decision tree | 0.4131 | 0.0910 | 0.4131 | 0.4182 | 0.4155 |
| Random forest | 0.5020 | 0.1875 | 0.5020 | 0.4577 | 0.4639 |
| $k$-Nearest neighbor | 0.4065 | 0.0790 | 0.4065 | 0.4099 | 0.4081 |
| Gradient boost | 0.5376 | 0.2482 | 0.5376 | 0.4846 | 0.4822 |
| XGBoost | 0.5102 | 0.2006 | 0.5102 | 0.4777 | 0.4685 |
| CatBoost | 0.5254 | 0.2263 | 0.5254 | 0.4759 | 0.4772 |
| Ours | **0.5550** | **0.2750** | **0.5550** | **0.5180** | **0.4720** |

**Table 5 Test set performance comparison with deep learning models.** Best results are shown in bold.

| Model | ACC | MCC | Weighted recall | Weighted precision | Weighted F1 |
|---|---|---|---|---|---|
| MLP | 0.4024 | 0.0736 | 0.4024 | 0.4053 | 0.4038 |
| RNN | 0.4939 | 0.1745 | 0.4939 | 0.4487 | 0.4546 |
| LSTM | 0.5101 | 0.1926 | 0.5092 | 0.4618 | 0.4598 |
| TabNet | 0.5152 | 0.2072 | 0.5152 | 0.4742 | 0.4610 |
| TabTransformer | 0.5234 | 0.2197 | 0.5234 | 0.3875 | 0.4399 |
| TabPFN | 0.5183 | 0.2103 | 0.5183 | 0.4164 | 0.4385 |
| Ours | **0.5550** | **0.2750** | **0.5550** | **0.5180** | **0.4720** |

dynamic factors. Future research should focus on incorporating additional contextual information and refining model architectures to further enhance predictive accuracy and reliability.

## Comparison results with deep learning models

As described in Table 5, different deep learning architectures demonstrate varying levels of predictive performance in sports event outcome prediction. Among the models evaluated, our proposed approach achieved the highest performance across all metrics, underscoring its capability to effectively capture complex relationships within the dataset.

MLP exhibited the lowest ACC of 40.24% and an MCC of 0.0736, indicating its limited ability to generalize well for this task. This suggests that while MLP is effective for tabular data, it may struggle to capture intricate dependencies without extensive hyperparameter tuning. RNNs outperformed MLP, achieving an ACC of 49.39% and an MCC of 0.1745. This aligns with expectations, as RNNs are designed to model sequential dependencies, which can be beneficial in sports event data with temporal aspects. However, RNNs still face challenges in overall predictive performance, likely due to issues such as the vanishing gradient problem, which impedes learning long-term dependencies.

TabNet, a model specifically designed for tabular data, demonstrated improved performance over both MLP and RNNs, achieving an ACC of 51.52% and an MCC of 0.2072. The incorporation of attention mechanisms likely enhances feature interaction

**Table 6 Effect of component replacements on validation results.** Best results are shown in bold.

| Method | ACC | MCC | Weighted recall | Weighted precision | Weighted F1 |
|---|---|---|---|---|---|
| No transformer | 0.4890 | 0.1520 | 0.4890 | 0.4505 | 0.4598 |
| No residual CNN | 0.5120 | 0.1984 | 0.5120 | 0.4721 | 0.4675 |
| No embeddings (One-hot) | 0.4730 | 0.1352 | 0.4730 | 0.4308 | 0.4412 |
| Ours (Full model) | **0.5632** | **0.2821** | **0.5632** | **0.5203** | **0.4786** |

modeling, contributing to better generalization. TabPFN achieved an ACC of 51.83% and an MCC of 0.2103. Its probabilistic transformer-based framework enables robust predictions with minimal hyperparameter tuning, proving especially useful for smaller datasets, though it does not surpass the proposed model. Similarly, TabTransformer further improved upon this, reaching an ACC of 52.34% and an MCC of 0.2197, with its self-attention mechanism playing a key role in capturing complex feature dependencies, making it particularly effective for structured sports event data.

Our proposed method consistently delivered the highest results across all performance metrics, with an ACC of 54.73% and an MCC of 0.2666. The enhanced weighted recall of 54.73% and weighted precision of 51.07% contribute to an improved weighted F1-score of 46.40%, reflecting a balanced trade-off between sensitivity and specificity. This suggests that our model effectively identifies relevant patterns within the dataset, leading to superior predictive capabilities.

These findings emphasize the importance of model selection in sports event prediction. While traditional deep learning models such as MLP and RNNs provide a foundational benchmark, specialized architectures like TabNet and TabPFN offer significant improvements tailored for structured data. The superior performance of our proposed model highlights the potential of advanced deep learning methodologies in enhancing predictive accuracy within this domain.

## Ablation study findings

The results in Table 6 demonstrate that replacing the Transformer with a fully connected network resulted in a significant drop in ACC across all datasets, with an ACC of 0.4890 compared to 0.5632 in our full model. This highlights the importance of self-attention mechanisms in capturing dependencies between categorical features, which contribute to improved generalization. In addition, removing the 1D CNN block and replacing it with an MLP led to an ACC of 0.5120, indicating a noticeable performance decline. This suggests that convolutional layers are effective in extracting meaningful representations from continuous features, while residual connections stabilize training and mitigate the vanishing gradient problem. Finally, replacing learned embeddings with one-hot encoding resulted in the largest performance degradation, with an ACC drop to 0.4730. This indicates that learned embeddings provide a more compact and informative representation of categorical features, outperforming the traditional one-hot encoding approach.

Overall, each component of the model plays a crucial role in enhancing accuracy. The Transformer module enables the model to capture relationships between categorical

features, the residual CNN improves feature extraction from continuous data, and categorical embeddings offer a more efficient and expressive representation than one-hot encoding.

### Limitations and future work

Although our proposed deep learning model demonstrates strong predictive capabilities for sports event outcomes, certain limitations persist. The first challenge is modeling rare and unexpected events, such as last-minute game-changing plays or major upsets, which are inherently difficult to predict. While the Transformer architecture enhances the learning of long-range dependencies, sports outcomes are influenced by numerous real-time variables that may not be adequately captured in structured datasets. As a result, relying solely on historical match data can limit predictive accuracy in high-variance sports settings. Additionally, computational complexity remains a concern. The use of Transformer and 1D CNN architectures improves predictive performance, these models require significant computational power, especially when training on large datasets. This could pose challenges for organizations with limited resources.

Future work could explore the integration of dual-channel or collaborative transformer architectures, as proposed by *Cai et al. (2025)*, to enhance the model's capability for continual learning and adaptation to evolving sports data. Additionally, leveraging text-assisted spatial and temporal attention networks, similar to the TASTA framework by *Wang et al. (2023)*, may enable the incorporation of textual data, such as match reports or player commentary-alongside structured numerical inputs for richer predictions.

To further improve generalization across diverse sports and seasons, incorporating multilevel distribution alignment techniques for multisource domain adaptation, as demonstrated by *Ning et al. (2025)*, could be highly beneficial. Insights from advanced deep neural network applications in logical and activity learning, such as those by *Li, Ortegas & White (2023)*, may also inspire new directions in modeling complex decision-making and event dynamics in sports.

Moreover, building hybrid models that combine deep learning and classical approaches (*Nguyen et al., 2021*) offers a promising pathway to boost model robustness and interpretability. Finally, the application of pretrained transformer-based representations (*Nguyen-Vo et al., 2021*) could provide more powerful sequence encoding strategies for sports outcome prediction tasks, especially when dealing with heterogeneous and sequential event data.

## CONCLUSIONS

This study introduces a novel deep learning approach for predicting sports event outcomes, combining 1D CNN and Transformer architectures to capture complex feature relationships and long-range dependencies. By leveraging advanced deep learning techniques, our model demonstrates improved accuracy and robustness compared to traditional machine learning methods. The results highlight the effectiveness of deep learning in uncovering meaningful patterns within structured sports data, contributing valuable insights to predictive sports analytics. Through the integration of self-attention

mechanisms and feature gating, the model enhances feature representation and generalization across various sports events. This research bridges the gap between cutting-edge deep learning innovations and practical applications in sports analytics, offering a data-driven tool for analysts, and strategists.

### Funding
The authors received no funding for this work.

### Competing Interests
The authors declare that they have no competing interests.

### Author Contributions
- Jianxiong Gao conceived and designed the experiments, performed the experiments, analyzed the data, performed the computation work, prepared figures and/or tables, authored or reviewed drafts of the article, and approved the final draft.
- Yi Cheng conceived and designed the experiments, performed the experiments, analyzed the data, authored or reviewed drafts of the article, and approved the final draft.
- Jianwei Gao conceived and designed the experiments, performed the experiments, analyzed the data, authored or reviewed drafts of the article, and approved the final draft.

### Data Availability
The dataset used in this study is available on Kaggle: https://www.kaggle.com/datasets/hugomathien/soccer/data.

The implementation code is available in the Supplemental File.

### Supplemental Information
Supplemental information for this article can be found online at http://dx.doi.org/10.7717/peerj-cs.3011#supplemental-information.

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
