# Peer review of "Predicting sport event outcomes using deep learning"

_PeerJ Computer Science, doi:10.7717/peerj-cs.3011_

## Round 0.1 · original submission · Major Revisions

Please consider the comments and suggestions by the two reviewers and revise the manuscript accordingly. Consider comparing the proposed method with other state-of-the-art models for tabular data, such as the TabTransformer model, as suggested by Reviewer 2. If there are any comments that you don't agree with, please provide your explanations.

**Language Note:** PeerJ staff have identified that the English language needs to be improved. When you prepare your next revision, please either (i) have a colleague who is proficient in English and familiar with the subject matter review your manuscript, or (ii) contact a professional editing service to review your manuscript. PeerJ can provide language editing services - you can contact us at [email protected] for pricing (be sure to provide your manuscript number and title). – PeerJ Staff

Reviewer 1 ·

Basic reporting

The introduction clearly explains the problem. Literature well references and relevant.

The experiments on time-series data are thorough, and the figures and tables are easy to understand.

However, some details are missing, and the comparison section could use more work. More details are as below.

Experimental design

One of the good points is that the paper gives a clear description of the proposed model, complete with detailed block diagrams, and the accompanying code and data are available.

The experimental setup is sound, but it would help to include training details, like the number of epochs and learning rate, etc.

The authors mentioned using a benchmark sports dataset, but the class sizes were not specified. What are the counts for the training, validation, and test sets?

The authors may want to provide more details about the method to encode categorical data before passing it into the transformer.

Validity of the findings

The experiments and evaluations performed satisfactorily.

The paper’s future research section needs more detailed ideas. For example, the authors could look into how well the model scales to real-time predictions, like during live sports events.

Cite this review as
Anonymous Reviewer (2025) Peer Review #1 of "Predicting sport event outcomes using deep learning (v0.1)". PeerJ Computer Science

Reviewer 2 ·

Basic reporting

Authors propose a hybrid deep learning architecture that integrates a 1D CNN for handling continuous features and a Transformer module for processing categorical data embeddings for predicting outcomes of sports events.
● The paper is well-structured and provides a clear and coherent introduction that effectively sets the stage for the study.
● The motivation is well-stated, and the relevance of using both CNN and Transformer components is justified in the context of mixed-type input data.
● The experimental results are reported clearly, with meaningful comparisons and sufficient baseline models for benchmarking.
● Clear and unambiguous, professional English used throughout.

Experimental design

● The Data Processing section is clearly described and easy to follow. The visualization of variable distributions adds to the interpretability of the input data. However, the dataset is divided into three labels: win, draw, and defeat, but the distribution of samples across these classes is not provided. Including this information would enhance the understanding of class balance.
● For the time series data, LSTM models are generally considered strong baselines. It would be helpful if the author could include comparisons with a LSTM-based model to support the experimental results.
● In the Hyperparameter Tuning section, the focus seems to be primarily on parameters for traditional machine learning models. The author should provide more details on the hyperparameters used for the deep learning models as well.

Validity of the findings

● The results show clear improvements over both traditional machine learning and deep learning approaches. However, it would further strengthen the work to include comparisons with recent state-of-the-art architectures designed for tabular data, such as TabTransformer.

Cite this review as
Anonymous Reviewer (2025) Peer Review #2 of "Predicting sport event outcomes using deep learning (v0.1)". PeerJ Computer Science

---

## Round 0.2 · accepted · Accept

The authors have addressed all of the reviewers' comments. Based on their recommendations and my own assessment, the manuscript is ready for publication.

Reviewer 1 ·

Basic reporting

The revised manuscript meets the standards. No further comments.

Experimental design

The revised manuscript has been improved, and it meets the standards. No further comments.

Validity of the findings

No further comments.

Additional comments

The authors have revised the manuscript significantly. I have no further comments. The manuscript can be accepted as it is.

Cite this review as
Anonymous Reviewer (2025) Peer Review #1 of "Predicting sport event outcomes using deep learning (v0.2)". PeerJ Computer Science

Reviewer 2 ·

Basic reporting

- The manuscript is clear and well-structured, effectively presenting the motivation for the hybrid deep learning architecture.
- The use of professional English makes the content clear and unambiguous.

Experimental design

- The addition of class distribution information provides improved clarity regarding class balance.
- Adding comparisons with an LSTM-based model and additional details on deep learning hyperparameters has improved the completeness of the experimental design.

Validity of the findings

- Results now include comparisons with state-of-the-art architectures such as TabTransformer, addressing prior suggestions.
- The expanded set of experimental results and baselines supports the validity and effectiveness of the proposed approach.

Cite this review as
Anonymous Reviewer (2025) Peer Review #2 of "Predicting sport event outcomes using deep learning (v0.2)". PeerJ Computer Science